

# Biotea: semantics for Pubmed Central

Alexander Garcia[1], Federico Lopez[2], Leyla Garcia[3], Olga Giraldo[1],
Victor Bucheli[2] and Michel Dumontier[4]

[1] Ontology Engineering Group, Universidad Politécnica de Madrid, Madrid, Spain
[2] Escuela de Ingeniería de Sistemas y Computación, Universidad del Valle, Cali, Colombia
[3] Temporal Knowledge Bases Group, Department of Computer Languages and Systems, Universitat Jaume I, Castelló de la Plana, Spain
[4] Maastricht University, Institute of Data Science, Maastricht, The Netherlands

## ABSTRACT

A significant portion of biomedical literature is represented in a manner that makes it difficult for consumers to find or aggregate content through a computational query. One approach to facilitate reuse of the scientific literature is to structure this information as linked data using standardized web technologies. In this paper we present the second version of Biotea, a semantic, linked data version of the open-access subset of PubMed Central that has been enhanced with specialized annotation pipelines that uses existing infrastructure from the National Center for Biomedical Ontology. We expose our models, services, software and datasets. Our infrastructure enables manual and semi-automatic annotation, resulting data are represented as RDF-based linked data and can be readily queried using the SPARQL query language. We illustrate the utility of our system with several use cases. Our datasets, methods and techniques are available at http://biotea.github.io.

Corresponding author
Alexander Garcia,
alexgarciac@gmail.com

## BACKGROUND

Semantic publishing (*Shotton, 2009*; *Shotton et al., 2009*) has been defined as the *enhancement of scholarly publications by the use of modern web standards to improve interactivity, openness and usability, including the use of ontologies to encode rich semantics in the form of machine-readable Resource Description Framework (RDF) metadata* (*Shotton & Peroni, 2016*; *RDF Working Group, 2014*). Publishers are actively enriching their content with semantics and generating machine-processable publications; for instance, Springer-Nature has released SciGraph (http://scigraph.com; *Springer Nature, 2017*), which is their linked data platform for allowing users to search in a more flexible way. Currently, it brings together data on roughly 8,000 proceedings volumes from around 1,200 conference series, including Springer's Lecture Notes in Computer Science (LNCS) (*Springer, 2015*). The Cochrane society is also working on a linked data platform (*Cochrane, 2017*); they are focusing in the characterization of the Population, Intervention, Comparison, Outcome (PICO) model (*Huang, Lin & Demner-Fushman, 2006*). Both efforts illustrate business models built upon the concept of data as a service; they are also a response to the need for more flexible ways to process scientific content going beyond presenting HTML and PDFs over index based query systems.

In this paper we present Biotea, our contribution to semantic publishing. In the Biotea project, we have semantically represented and annotated the full-text open-access subset of PubMed Central (PMC) (*NCBI, 2017c*); this subset currently includes articles from 7,407 journals. PMC is a free full-text archive of biomedical literature; articles under its open-access subset (PMC-OA) are still protected by copyright but are also available under the Creative Commons license; thus, a more liberal redistribution is allowed. We are extracting structured information from articles in PubMed Central and modeling it with general purpose bibliographic ontologies as well as with controlled vocabularies representing sections in combination with biomedical ontologies to semantically represent and annotate the literature. We are reusing existing ontologies in order to represent, title, authors, journal, sections, subsections and paragraphs and, the domain knowledge, e.g., diseases, chemical compounds, reagents, drugs, etc. We identify meaningful elements, e.g., biomolecules, chemical reagents, drugs, diseases, and other biomedical entities, within the content and represent these as semantic annotations. The annotations are associated to well-known biomedical ontologies. Biotea aims to aggregate annotations from different pipelines and have them under a common representation, that of the Annotation Ontology (AO) (*Ciccarese et al., 2011*) or the Open Annotation Data Model (OADM) (*Sanderson & Ciccarese, 2013*). The provenance of the annotations is fully identified in our model; thus, making it possible to retrieve annotations from a specific user, in the case of human annotations or, from a specific annotation pipeline. Currently, we are only working with annotations from the National Center for Biomedical Ontologies (NCBO) annotator (*Jonquet, Shah & Musen, 2009*) as well as with human annotations; future versions of the dataset will include other annotation pipelines.

Semantic annotations and linked data technology make it possible to use ontology concepts to formulate queries; thus, retrieving papers about "*calcitonin and kidney injury together with Uniprot proteins that have calcitonin binding as molecular function as well as the calcitonin resource description from DBPedia (Bizer et al., 2009)*" is possible. The queries can easily be expanded by adding concepts and data sources. The biomedical linked data infrastructure facilitates to expand the query by indicating data sources capable of resolving specific parts of it; this is supported by the SPARQL specification (*SPARQL Working Group, 2013*). Semantic annotations also make it possible to compare sections from different papers with respect to one or more ontologies, e.g., "*what chemical entities do papers have in common in the Methods sectio*". Our model facilitates making granular queries focusing on entities in specific sections; for instance, it allows us to retrieve papers mentioning "*FTR and bronchial epithelial cell in the Results section*".

Our approach addresses a post publication problem; published papers are primarily available as HTML and PDF making little use of the available linked data infrastructure. Moreover, published content is not part of the linked data cloud; bibliographic metadata has been privileged over full content. We make it possible to expose the content in a format that is more amicable for machines to process and native to the semantic web. The papers that we are transforming to RDF have been published and deposited in PubMed Central, they are available as Journal Article Tag Suit files (JATS/XML) (*NISO, 1995*; *U.S. National Library of Medicine, 2017a*). JATS is an industry standard commonly used

in publication workflows. Our methods and techniques could easily be applied to any publication workflow producing JATS/XML. Throughout this paper we use RDFize as a verb, meaning (i) to generate an RDF representation of something that was originally in a different format and (ii) to convert or transform to RDF. We are RDFizing the corpus of documents, annotating it with biomedical ontologies and exposing the resulting dataset as linked open data.

This second version of Biotea is based on our previous work (*Garcia Castro, McLaughlin & Garcia, 2013*) and advances the state of the art in the following way: (i) it delivers a modularized process for generating RDF in order to make it more manageable—see sections ''The Publication Parsing Process'' and ''The Semantic Enrichment Process'' under ''Materials and Methods'' for more information; (ii) it makes it possible to generate annotations based on the Open Annotation Data Model (*Sanderson & Ciccarese, 2013*) in addition to Annotation Ontology (*Ciccarese et al., 2011*) that was supported by the first version of this work—see ''The Semantic Enrichment Process'' under ''Materials and Methods'' as well as, ''Semantically Enriched Content'' under the ''Results'' section; (iii) the model has been simplified by removing ontologies that are no longer in use, e.g., CNT (*Koch, Velasco & Ackermann, 2011*), see the ''Results'' section for a description of the model; (iv) the representation of publishers and provenance has also been modified and; (v) we have added support for human annotations via http://hypothes.is (*Hypothesis Project, 2017*), see ''Supporting Human Annotation'' under the ''Results'' section. The *Hypothesis Project (2017)* is an annotation platform that makes it possible for end users to easily annotate and share annotations for specific parts within the document. Our current stack of software makes it easier to add other annotation pipelines; in this way the corpus of annotations can be extended and made more specific, e.g., by adding protein-protein interactions annotation pipelines. We present examples illustrating the use of our dataset in the ''Using Biotea'' section.

## MATERIALS AND METHODS

The overall RDFization process has two main sub processes, namely, the Publication Parsing and Semantic Enrichment processes. The Publication Parsing RDFizes metadata, references, structure and content (*Biotea, 2017i*) while the Semantic Enrichment process uses Named Entity Recognition (NER) systems to identify expressions and terminology related to biomedical ontologies that are then RDFized as annotations (*Biotea, 2017i*). The Biotea projects are all MAVEN projects so dependencies are downloaded automatically; the software is available at https://github.com/biotea (JAVA 1.8 is required). We recommend to build and run using any Integrated Development Environment (IDE); we have used Eclipse Luna and Eclipse Neon. We tested the software in Ubuntu, Mac OSX Sierra 10.4 and Windows 7. We are also providing JAR files, further details about usage and parameters together with some examples are provided in the corresponding GitHub repositories. More information about the software, how to use it and latest versions can be found at https://github.com/biotea; information about the docker container is available at http://biotea.github.io/software/.
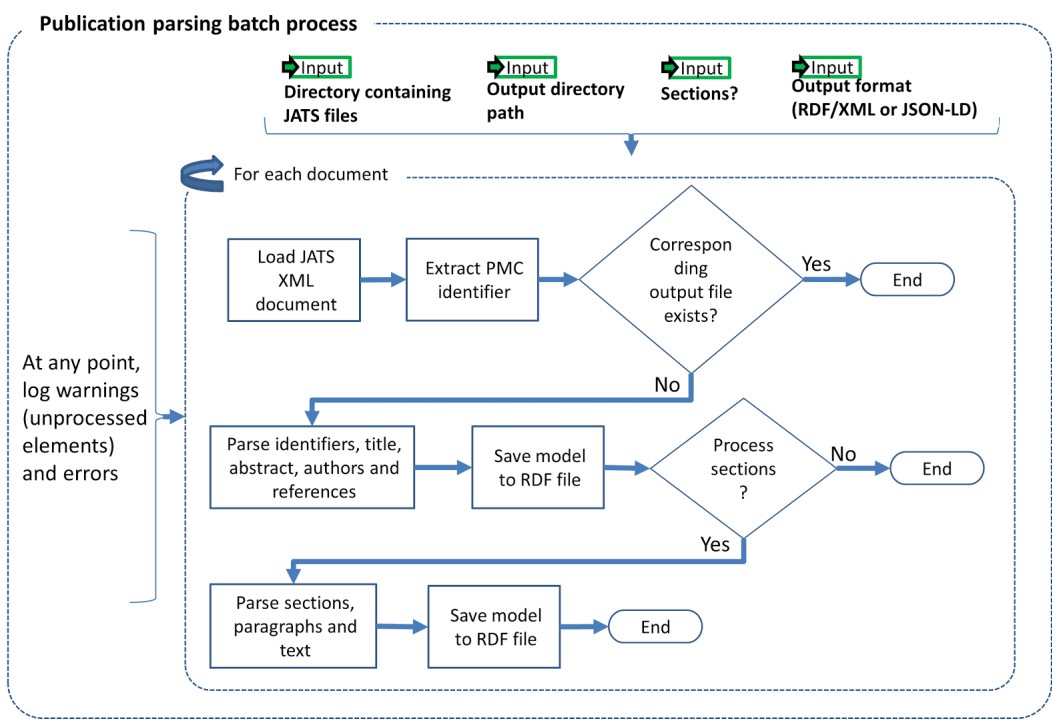

**Figure 1** Publication parsing process.

## The publication parsing process

The input to our Publication Parsing process are the articles from PMC-OA (*NCBI, 2017b*) in the Journal Articles Suite (JATS) format (*U.S. National Library of Medicine, 2017a*), i.e., XML files following a specific meta model. Our RDFization entails generating one RDF to represent the metadata and references and another one to represent the structure—sections and paragraphs, and content—actual text. Figure 1 illustrates the Publication Parsing process. We are representing sections, e.g., material and methods, as well as structural elements, e.g., citations, authors, of the paper. The Publication Parsing process brings together several ontologies into the Biotea model, see Table 1 for a detailed description of the ontologies. We are using BIBO (*D'Arcus & Giasson, 2009*), DoCO (*Constantin et al., 2016*) and Dublin Core Terms (DCTERMS) (*DCMI Usage Board, 2012*) to represent the structure of the document. For instance, a set of sections is represented as several `doco:Section` elements aggregated in a section list (`rdf:Seq`) that keeps the order as defined in the input JATS/XML document. The hierarchical structure amongst the section list, the sections and the subsections is represented using the `dcterms:hasPart` property.

## The semantic enrichment processes

We identify and annotate meaningful fragments within paragraphs by using the NER service provided by the NCBO Annotator. The NCBO Annotator (*Jonquet, Shah & Musen, 2009*; *NCBI, 2017a*) is part of the BioPortal platform (*Whetzel et al., 2011*); it provides

**Table 1  Ontologies used for metadata, structure, content and references.**

| Ontology | Purpose | Main elements used in Biotea |
|---|---|---|
| Bibliographic ontology (*D'Arcus & Giasson, 2009*) | Metadata | bibo:AcademicArticle, bibo:Document, bibo:doi, bibo:identifier, bibo:issn, bibo:Issue, bibo:issue, bibo:Journal, bibo:numPages, bibo:pageEnd, bibo:pageStart, bibo:pmid, bibo:shortDescription, bibo:volume |
| | References | bibo:AcademicArticle, bibo:Book, ibo:Chapter, bibo:citedBy, bibo:cites bibo:Document, bibo:Proceedings |
| Biotea (*Garcia Castro, McLaughlin & Garcia, 2013*) | Metadata (list of elements) | biotea:authorList |
| | Structure (list of elements) | biotea:paragraphList, biotea:sectionList |
| Document ontology (*Constantin et al., 2016*) | Structure and content | doco:Figure, doco:Section, doco:Paragraph, doco:Table |
| Dublin core terms (*DCMI Usage Board, 2012*) | Metadata | dcterms:description, dcterms:issued, dcterms:publisher, dcterms:title |
| | Provenance | dcterms:creator, dcterms:hasFormat, dcterms:isFormatOf, dcterms:references, dcterms:source |
| Friend of a friend ontology (*Brickley & Miller, 2014*) | Metadata | foaf:familyName, foaf:givenName, foaf:name, foaf:OnlineAccount, foaf:Organization, foaf:Person, foaf:publications |
| | References | foaf:familyName, foaf:givenName, foaf:name, foaf:OnlineAccount, foaf:Organization, foaf:Person, foaf:publications |
| OWL (*OWL Working Group, 2012*) | Link to other semantic representations | owl:sameAs |
| Provenance ontology (*Belhajjame et al., 2013*) | Provenance | prov:generatedAtTime, prov:wasAttributedTo, prov:wasDerivedFrom |
| RDF (*RDF Working Group, 2014*) | Content (text in paragraphs) | rdf:value |
| RDFS (*RDFS Working Group, 2014*) | Link to related web pages | rdfs:seeAlso |
| Semantic science integrated ontology (*Dumontier et al., 2014*) | Provenance | sio:is_data_item_in |

access to more than 350 ontologies and terminologies. The NCBO annotator makes it possible to semantically annotate text by recognizing the entities and establishing a link to an ontology; these annotations are often used to compose queries that bring together data elements from different resources. The NCBO Annotator is based on Mgrep (*Dai et al., 2008*); it recognizes and associates expressions in the text with unique concepts from biomedical ontologies. The NCBO Annotator utilizes to its advantage the hierarchy in the vocabularies used for the association. The annotation process is illustrated in Fig. 2.

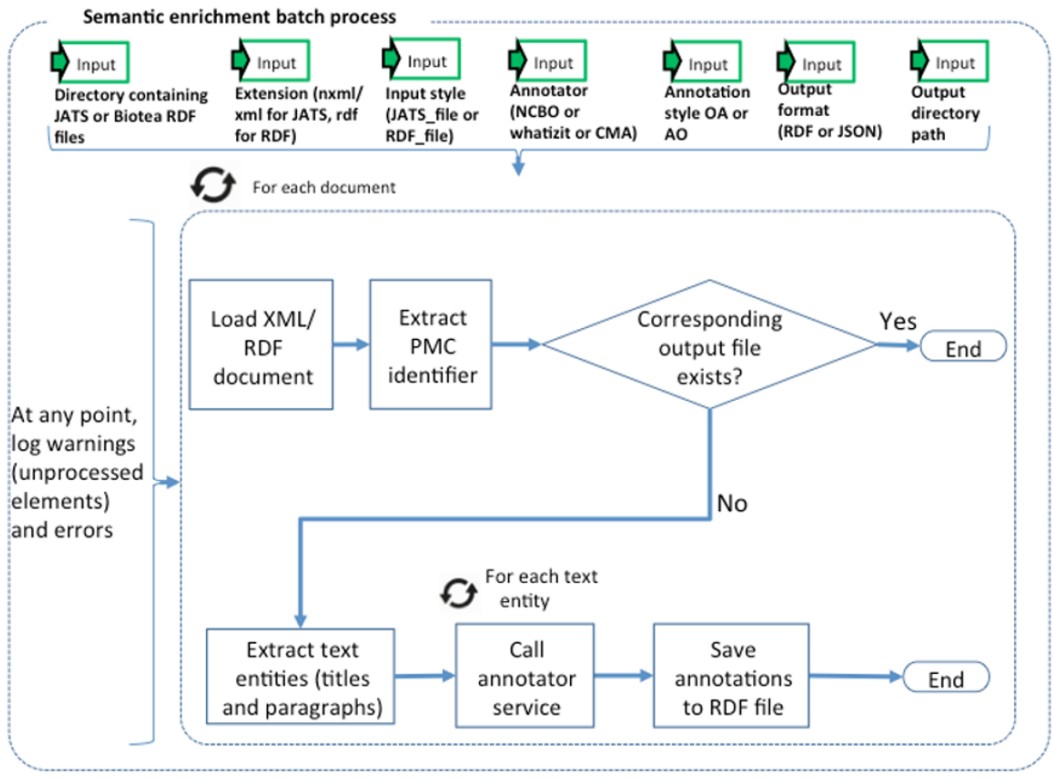

**Figure 2  Semantic enrichment.**

We are representing the identified entities by using either the Annotation Ontology (AO) (*Ciccarese et al., 2011*) or the Open Annotation Data Model (OADM) (*Sanderson & Ciccarese, 2013*). These annotation ontologies are used to semantically represent the annotations coming from the annotator as well as their links to ontological concepts in biomedical vocabularies. In this way, for each PMC article, we are generating RDF representing the structure and domain knowledge. The ontologies used for annotating are listed in Table 2.

The methods that we have developed for annotating allow parameterization. The users define the ontologies to be used, the list of stop words, the URL of service instance to use and the output format (AO or OADM, RDF-XML or JSON-LD). In addition, users can also parametrize what parts of an article to annotate, e.g., titles and abstracts only or full text.

## RESULTS

Our RDF data model follows the principles proposed by Tim Berners-Lee for publishing Linked Data (*Berners-Lee, 2006*), namely: (i) using Uniform Resource Identifiers (URIs) to identify things, (ii) using Hypertext Transfer Protocol (HTTP) URIs to enable things to be referenced and looked up by software agents, (iii) representing things in RDF and providing a SPARQL endpoint, and (iv) providing links to external URIs in order to
**Table 2  Ontologies used to support the annotation process.**

| Ontology | Purpose | Main elements used in Biotea |
|---|---|---|
| Annotation ontology (*Ciccarese et al., 2011*) | Annotation | ao:Annotation, ot:ExactQualifier, ao:body |
| | Link to biomedical ontologies | ao:hasTopic |
| | Link to RDFized publication | ao:annotatesResource, ao:context, ao:onResource |
| Biotea (*Garcia Castro, McLaughlin & Garcia, 2013*) | Frequency (occurrences and inverse document frequency) | biotea:idf, biotea:tf |
| Open Annotation Data Model (*Sanderson & Ciccarese, 2013*) | Annotation | oa:Annotation, oa:hasBody (with a oa:TextualBody) |
| | Link to biomedical ontologied | oa:hasBody (with a direct link to the ontological concept) |
| | Link to RDFized publication | oa:hasSource, oa:hasTarget |
| Provenance, authoring and versioning ontology (*Ciccarese & Soiland-Reyes, 2013*) | Provenance | pav:authoredBy, pav:createdBy |
| Provenance ontology (*Belhajjame et al., 2013*) | Provenance | prov:generatedAtTime |

facilitate knowledge discovery. The resulting dataset is available at *Biotea (2017b)*. Our dataset comprises 1,623,541 articles from PMC, distributed across 7,407 journals. We are modeling relations to other resources representing the same entity as `owl:sameAs`; we link to the same article in the Bio2RDF PubMed dataset, the Document Object Identifier (DOI), and the Identifiers.org (http://identifiers.org; *Juty, Le Novere & Laibe, 2012*) representation. Relations to web pages are included as `rdfs:seeAlso`; we also include links to the article in the PubMed repository and the information service of identifiers.org. An example is provided in the following RDF/XML excerpt corresponding to the RDFization of the article "An Improved Protocol for Intact Chloroplasts and cpDNA Isolation in Conifers" (*Vieira et al., 2014*). The Biotea RDFized version is linked via `owl:sameAs` to Bio2RDF, identifiers.org and DOI, all of them providing versions of the corresponding article in PubMed.

```
<bibo:AcademicArticle rdf:about="http://linkingdata.io/pmcdoc/pmc/3879346">
  <owl:sameAs rdf:resource="http://bio2rdf.org/pubmed:24392157"/>
  <owl:sameAs rdf:resource="http://identifiers.org/pubmed/24392157"/>
  <owl:sameAs rdf:resource="http://dx.doi.org/10.1371/journal.pone.0084792"/>
  <rdfs:seeAlso rdf:resource="http://info.identifiers.org/pubmed/24392157"/>
  <rdfs:seeAlso rdf:resource="http://www.ncbi.nlm.nih.gov/pubmed/24392157"/>
</bibo:AcademicArticle>
```

Listing 1: RDF Example

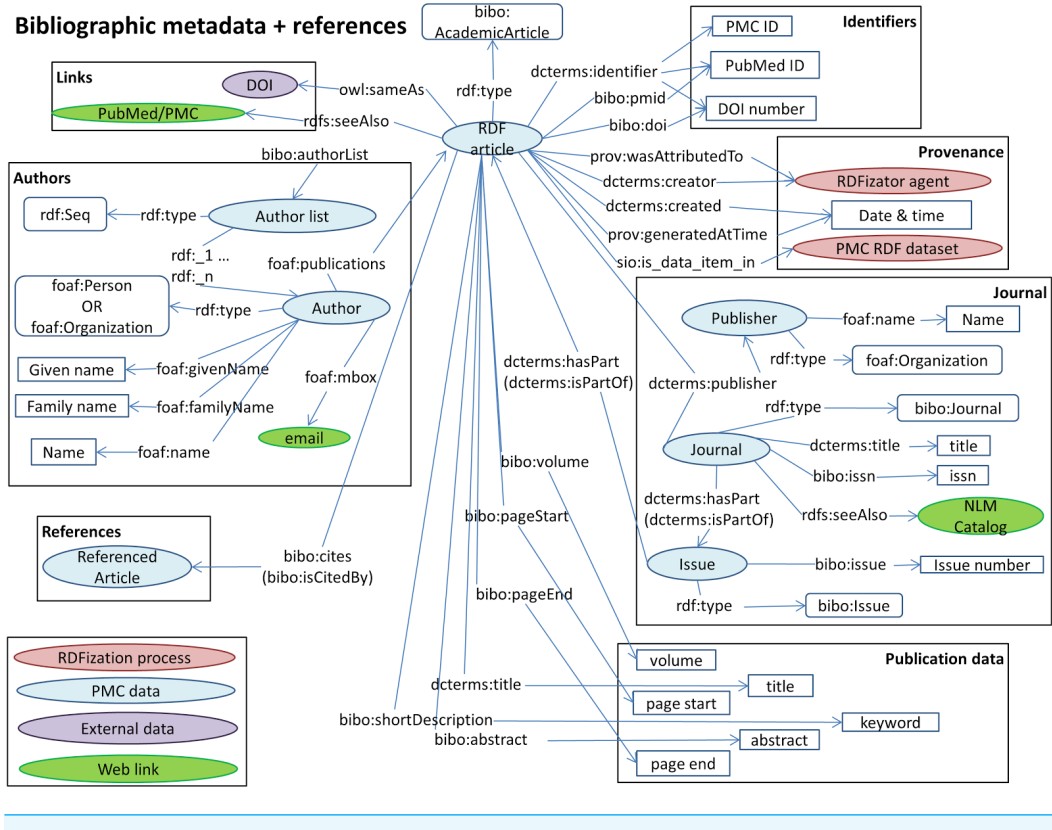

**Figure 3    The Biotea model.**

A general overview of our model is presented in Fig. 3. Our model describes identifiers, publication data, links, provenance, authors, references and sections. These are the structural elements of scientific papers.

We are using DOIs and PubMed IDs as identifiers for the articles. We use DCTERMS to represent titles and keywords. The abstracts are represented as BIBO elements, `bibo:abstract`. Authors are represented as a `bibo:authorList`; we use FOAF (*Brickley & Miller, 2014*) to fully represent authors, e.g., `foaf:givenName`, `foaf:mbox`. Authors may also be organizations, `foaf:Person`, `foaf:Organization`. By using these data elements we can support queries such as "retrieve the papers from PlosOne with Shun-Fa Yang as an author" or, "retrieve the DOIs authored by Shun-Fa Yang". The graph for sections and paragraphs is illustrated in Fig. 4. Sections include a title and a sequence of paragraphs modeled as `doco:Paragraphs`; the actual text is modeled as rdf:value. References include meta-data similar to that of the main article. This granularity in the representation of sections makes it possible to focus on specifics within sections; thus, retrieving "materials and methods using chloroplast DNA isolation methods" can be processed by the query illustrated below.

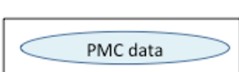

**Structure & Content**

**Only structure in RDF**
**(for those articles whose**
**full-text cannot be exposed)**

RDF article

Section list — rdf:type → rdf:Seq

rdf:_1 ... rdf:_n

dcterms:hasPart (dcterms:isPartOf)

dcterms:hasPart (dcterms:isPartOf) → subsection

rdf:Seq ← rdf:type — Section list

rdf:_1 ... rdf:_n

rdf:type

title ← dcterms:title — section

doco:Section

**References**

Referenced Article

bibo:cites (bibo:isCitedBy)

rdf:Seq ← rdf:type — Paragraph list

**Full-text in RDF**
**(for those articles with**
**full-text)**

rdf:_1 ... rdf:_n

dcterms:hasPart (dcterms:isPart Of)

Plain text ← rdf:value

doco:Paragraph ← rdf:type — paragraph

PMC data

**Figure 4   Text structure RDF model.**

```
PREFIX doco: <http://purl.org/spar/doco/>
PREFIX dcterms: <http://purl.org/dc/terms/>
PREFIX oa: <http://www.w3.org/ns/oa#>

SELECT ?content
{
  ?annotationChloroplastDNA a oa:Annotation .
  ?annotationChloroplastDNA oa:hasBody ?bodyChloroplastDNA .
  ?bodyChloroplastDNA rdf:value "Chloroplast DNA" .

  ?annotationIsolation a oa:Annotation .
  ?annotationIsolation oa:hasBody ?bodyIsolation .
  ?bodyIsolation rdf:value "Isolation" .

  ?annotationChloroplastDNA oa:hasTarget ?paragraph .
  ?annotationIsolation oa:hasTarget ?paragraph .
  ?section dcterms:hasPart ?paragraph .
  ?section dcterms:title "Materials and Methods" .
  ?paragraph rdf:value ?content .
}
```

Listing 2: SPARQL query

The positions within the text in the resulting RDF files vary depending on the input, these are different from those in the corresponding HTML or PDF. We are localizing the annotations with respect to the RDFized paragraph rather than to the original positions. In this way it is easier to query for annotations within the same paragraph or section. In order to select an RDF element, we use the class ElementSelector as defined in the Biotea

Ontology (*Biotea, 2017g*); this class is used as a domain for `ao:onResource` and as range for `ao:context`, the excerpt of code below illustrates this. Context identification is only required in AO. The OADM provides a simpler model where the publication, section or paragraph are linked via `oa:hasTarget`.

```
<aot:ExactQualifier rdf:about="http://bio2rdf.org/pmc_resource:annotationNCBO_1">
  <ao:annotatesResource rdf:resource="http://bio2rdf.org/pmc:3879346"/>
  <ao:context>
    <biotea:ElementSelector rdf:about="http://bio2rdf.org/pmc_resource:selector_1">
      <dcterms:references
rdf:resource="http://bio2rdf.org/pmc_resource:3879346_paragraph_Introduction_para_1"/>
      <ao:onResource rdf:resource="http://bio2rdf.org/pmc:3879346"/>
    </biotea:ElementSelector>
  </ao:context>
  <ao:body rdf:datatype="http://www.w3.org/2001/XMLSchema#string">GENES</ao:body>
</aot:ExactQualifier>
```

Listing 3: Using RDF element selectors in AO annotations

## Semantic enrichment

Our current implementation makes it possible to express the annotations generated by the NCBO Annotator using either the AO or the OADM. Figure 5 illustrates an example expressing the annotation in the OADM model; this is the default annotation ontology used in our RDFization process. In both cases we are making explicit the relation between the annotation and the location, e.g., section and document identifier; thus, making it possible to limit the query for an entity in a specific section of a document. We are using 20 domain ontologies from Bioportal to support the annotation, the ontologies are listed at (*Biotea, 2017c*).

## Supporting human annotation

We are now supporting human annotations coming from Hyphothesis (http://hypothes.is; *Hypothesis Project, 2017*). Hyphothesis is an open source web based annotation platform; it allows us to annotate PDFs as well as HTML. We have integrated http://hypothes.is into the LENS Reader interface (*Schekman, Watt & Weigel, 2013*); this user interface makes it possible for us to load JATS/XML from the PMC collection of documents and render it as HTML. The integration between Hypothesis and LENS delivers a unified user experience (UX); researchers load the integrated interface, log in the annotator and then annotation is a simple process of selecting text and annotating. Annotations coming from our instance of Hypothesis become part of the annotation cloud for the document via an identifier, e.g., DOI or PMC. The annotator is modeled as a `foaf:Person` who has a `foaf:mbox`. We are currently supporting only annotations from predefined vocabularies; Fig. 6 illustrates the interface, an online demo with LENS and Hyphothesis is available at *Biotea (2017f)*.

## Integration with Bio2RDF

Bio2RDF (*Belleau et al., 2008*) makes biomedical data available by using Semantic Web technologies such as RDF and SPARQL. Bio2RDF brings together information from diverse public databases such as DrugBank (*Wishart et al., 2006*; *Law et al., 2014*), MeSH (*Rogers, 1963*) and OMIM (*Amberger et al., 2015*) amongst others. Bio2RDF does not just provide

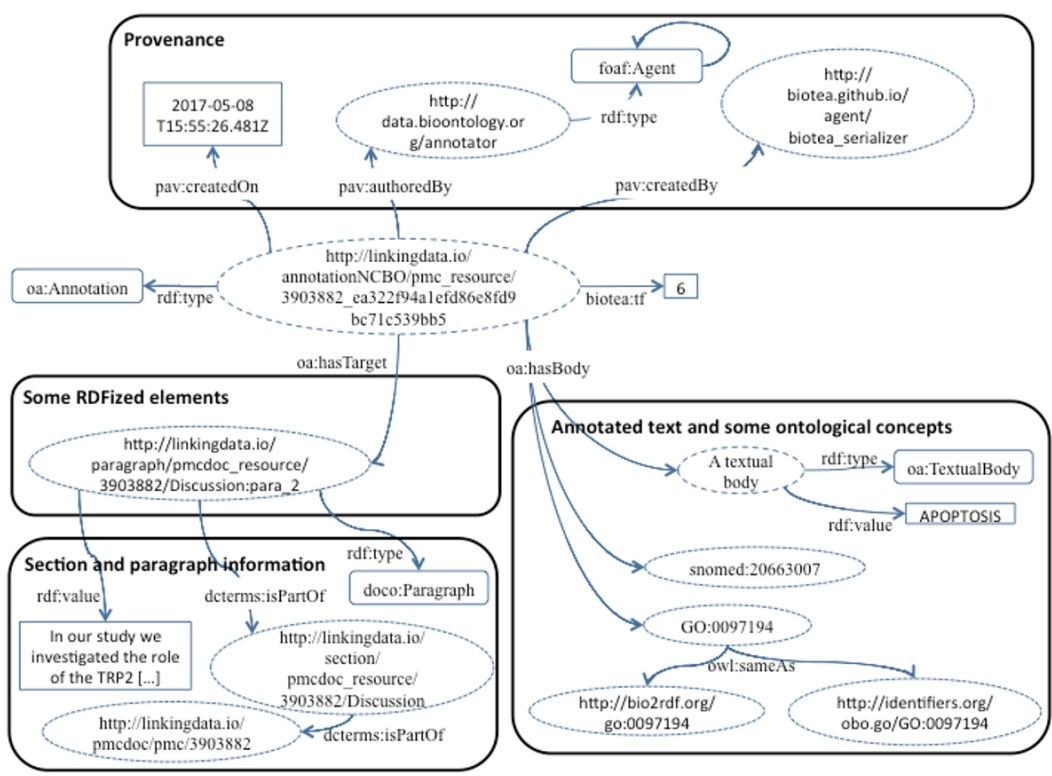

**Figure 5    Annotations based on the OADM model.**

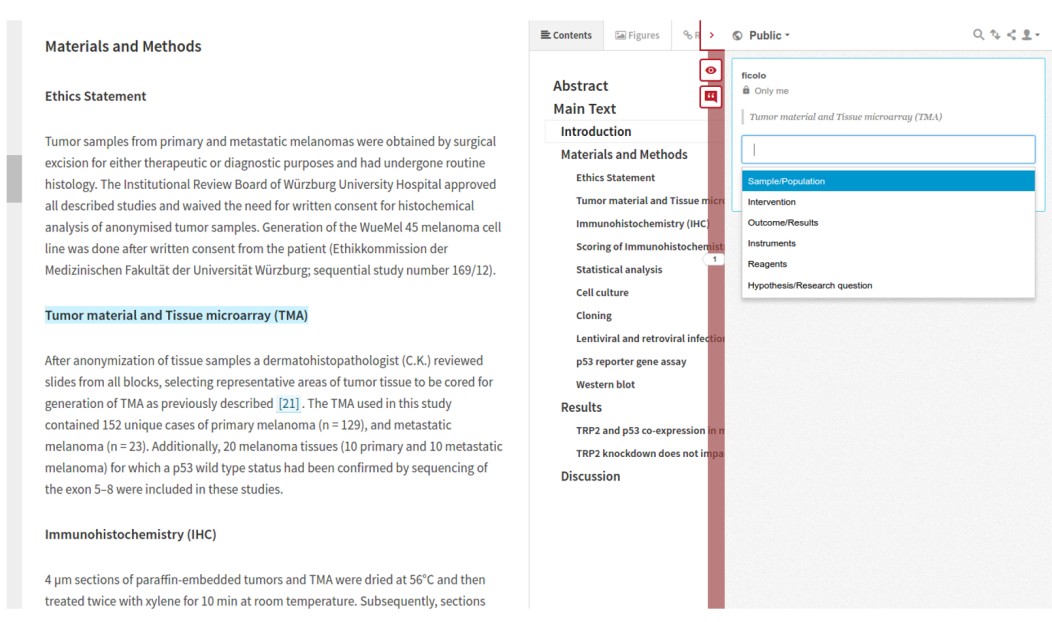

**Figure 6    Human annotation interface.**

a single entry point for all of these resources; it also transforms them into a common data model based on the Semantic science Integrated Ontology (SIO) (*Dumontier et al., 2014*). Our semantically enriched information layer for PMC articles, i.e., annotated content, makes extensive use of biomedical ontologies in similar ways to those in Bio2RDF. Having SIO compliant annotations simplifies the process of relating both datasets; our mappings address metadata, structural elements in the paper, content and, annotations.

We provide a mapping file for Bio2RDF in the form of a Java properties file. Classes and object properties from Biotea are mapped to SIO concepts, see *Biotea (2017g)*. For instance, the class `bibo:AcademicArticle` is mapped to `sio:peer-reviewed-article`, the object property bibo:cites is mapped to `sio:cites`. SIO only has one data type property—`sio:has-value`; in order to map datatype properties from Biotea to SIO we are converting these properties to object properties and then linking them to the most appropriate class depending on the mapping at hand. In this way we are encapsulating the original data type property value; thus, a `bibo:pmid` with the value "28300141" is mapped to the object property `sio:has-identifier`, this is linked to the class `sio:identifier` that is related by means of `sio:has-value` to the actual PMID "pmid:28300141".

Defining mappings to other models is also possible. In order to do so, a new Java property file has to be defined; in this file, the mappings will indicate the relations to elements in the Biotea model. The Bio2RDF mapping file can be used as a template for generating other mappings. The 1-to-1 nature of our mapping process poses a limitation; if a model has two classes to represent patents, e.g., `a_model:scientificPatent` and `a_model:industrialPatent`, then `bibo:Patent` will be mapped to only one of them. Such scenarios require adjustments in the ontology, BIBO in this case, being used by Biotea.

## Using Biotea

In our first experience with Biotea we explored the use of annotations as part of Graphical User Interfaces (GUIs). We built a simple prototype that facilitated the conceptual exploration of a paper via available annotations; the user could position the mouse over a cloud of annotations and then interactively see the text in which the annotation is located (*García-Castro, Castro & Gómez, 2012*). For this new release, we are searching over the dataset by establishing filters based on ontologies and then, visualizing and exploring the similarity of the resulting dataset. Initially, the dataset is filtered based on the selection of ontological concepts; these concepts belong to one or more of the ontologies used to annotate the dataset. For the resulting dataset, an ontology is selected for building the feature vector to be used as the basis for the clustering process. The final result indicates how closely related are the papers. The visualization is built upon a zoom-able dendogram that makes it easy for the end-user to explore the dataset and inspect the tree of similarity, this prototype is available at *Biotea (2017e)*.

Lets consider the following workflow, "retrieve papers annotated with the SNOMED CT term "American Joint Committee on Cancer" and then use SNOMED CT (*U.S. National Library of Medicine, 2017c*) to cluster the resulting dataset." We are using hierarchical agglomerative clustering with a complete linkage strategy using the cosine distance as metric for building the clusters. Figure 7 illustrates the resulting cluster.

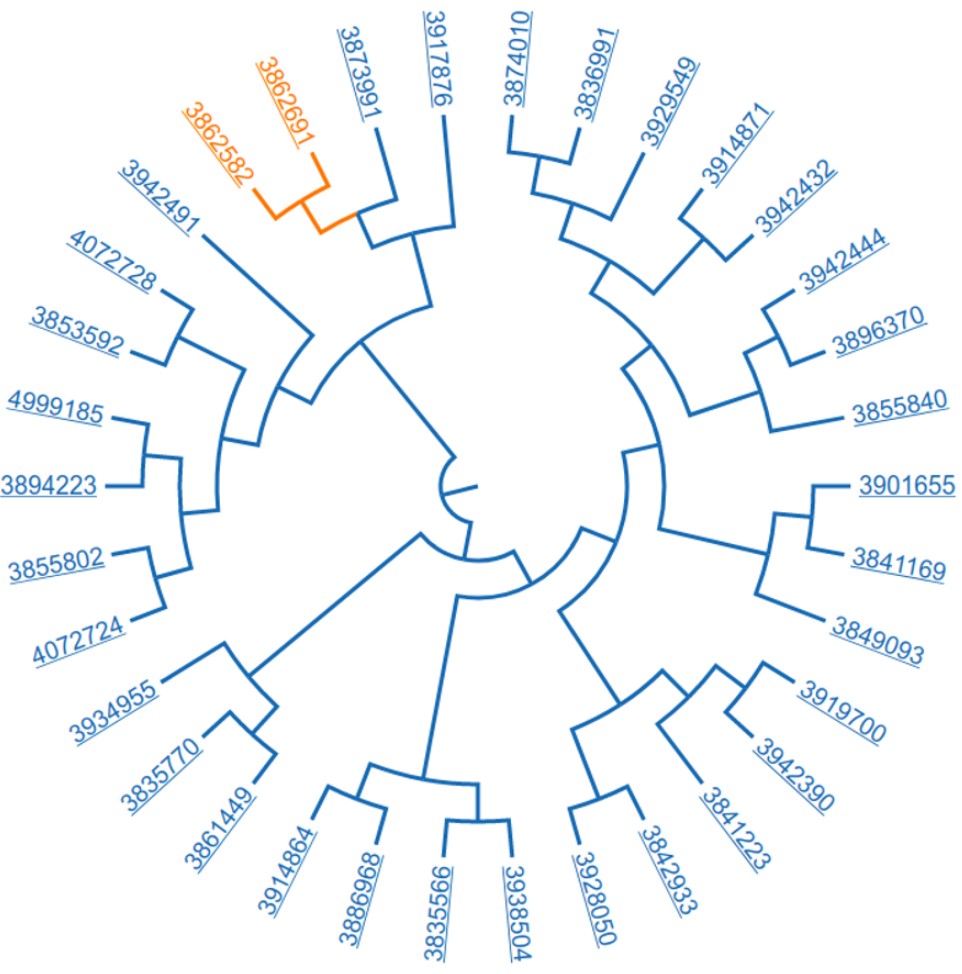

**Figure 7** Resulting dataset; 34 papers related "*American Joint Committee on Cancer*" and clustered based on SNOMED CT annotations.

We have manually analyzed the two papers that are closest to each other, see the first two rows in Table 3. We also analyzed one paper that is far apart from the first pair, see the last row in Table 3.

We found commonalities in the bibliographic information. The two related papers were published in the same date, December 13, 2013; they share one author, Shun-Fa Yang and he is affiliated to the Institute of Medicine, Chung Shan Medical University, Taichung, Taiwan. The commonalities across these papers also include:

1. Type of research: cancer. The SNOMED CT terms found in both papers, that helped us to identify that both articles are about cancer are listed in Table 4.

2. Patients studied:

   The patients are Taiwanese. Both papers addressed the consumption of tobacco. In addition, both papers report using the AJCC staging system; this is a classification system developed by the American Joint Committee on Cancer, hence the acronym, for describing the extent of disease progression in cancer patients (e.g., Tumor size,

**Table 3 Two PMC papers classified with a "middle similarity" and one paper with a distant similarity.**

| Document | PMCID | Title |
|---|---|---|
| doc1 (*Tsai et al., 2013*) | 3862691 | Impact of Interleukin-18 Polymorphisms -607A/C and -137G/C on Oral Cancer Occurrence and Clinical Progression |
| doc2 (*Wang et al., 2013*) | 3862582 | Impacts of CA9 Gene Polymorphisms on Urothelial Cell Carcinoma Susceptibility and Clinicopathologic Characteristics in Taiwan |
| doc3 (*Fan et al., 2014*) | 3942390 | The has-miR-526b Binding-Site *rs8506G > A* Polymorphism in the lincRNA-NR_024015 Exon Identified by GWASs Predispose to Non-Cardia Gastric Cancer Risk |

**Table 4 SNOMED CT terms related to cancer.**

| SNOMED CT term | ID |
|---|---|
| Carcinoma | snomedct:68453008 |
| Malignant neoplastic disease | snomedct:363346000 |
| Neoplasm | snomedct:108369006 |
| Neoplasm, malignant (primary) | snomedct:86049000 |

**Table 5 SNOMED CT terms describing the patients.**

| SNOMED CT term | ID |
|---|---|
| Tobacco user | snomedct:110483000 |
| Tobacco | snomedct:39953003 |
| Taiwanese | snomedct:63736003 |
| AJCC | snomedct:258236004 |

Lymph Nodes affected, Metastases). The SNOMED CT terms, related to the description of the patients are listed in Table 5.

3. Collecting and treating the sample:
   The type of sample collected from patients, treatment and storage conditions for the sample were the same: whole-blood placed in tubes containing ethylenediaminetetraacetic acid (EDTA), immediately centrifuged, and stored at −80 °C; see Table 6 for the corresponding SNOMED CT IDs.
4. Molecular methods used to identify the target genes:
   In order to find the associations between the gene of interest and predisposition to cancer, the authors used similar methods: (i) Genomic DNA extraction, (ii) Real-time PCR and (iii) Statistical analysis. The SNOMED CT terms found in both papers about the methods are listed in Table 7.

From the cluster presented in Fig. 7 we selected the paper, "The has-miR-526b Binding-Site rs8506G>A Polymorphism in the lincRNA-NR_024015 Exon Identified by GWASs Predispose to Non-Cardia Gastric Cancer Risk" (*Fan et al., 2014*), see third row, Table 3. It bears a weak relation with respect to those previously analyzed; this study provided evidence that genetic polymorphisms in the exonic regions of long intergenic noncoding

**Table 6  SNOMED CT terms related to the sample.**

| SNOMED CT term | ID |
| --- | --- |
| Whole blood | snomedct:420135007 |
| Ethylenediamine tetra-acetate | snomedct:69519002 |

**Table 7  SNOMED CT terms related to methods.**

| SNOMED term | ID |
| --- | --- |
| Probe with target amplification | snomedct:702675006 |
| Polymerase chain reaction | snomedct:258066000 |
| Deoxyribonucleic acid extraction technique | snomedct:702943006 |

RNAs (lincRNAs) play a role in mediating susceptibility to Non-Cardia Gastric Cancer (NCGC). The three papers share carcinoma. In addition, the tumor node metastasis (TNM) classification and tumor staging were evaluated in the three papers according to the American Joint Committee on Cancer Staging system; this is consistent with the initial query "retrieve papers annotated with the SNOMED CT term "American Joint Committee on Cancer". However, they differ significantly in the population, Taiwanese (doc1, 2) vs Chinese (doc 3). They also differ in the techniques; the doc 3 includes a SNP selection, genotyping analysis, cell culture, subcellular fractionation, construction of reporter plasmids, transient transfections and luciferase assays, expression vector construction, RNA isolation and Quantitative RT-PCR analysis and a cell visibility assay to demonstrate that the G to A base change at rs8506G>A disrupts the binding site for has-miR-526b, thereby influencing the transcriptional activity of lincRNA-NR_024015 and affecting cell proliferation.

### In and out the content, making use of Linked Data

Biotea makes it easy to integrate the literature, e.g., PubMed Central, into more complex queries. Table 8 presents sample queries, some of them making use of external resources -e.g., Uniprot. Our SPARQL endpoint is accessible at (*Biotea, 2017d*), all queries are available at *Biotea (2017h)*.

A researcher may be interested in the following workflow "retrieve all the pathways referencing "insulin" from Reactome (*Fabregat et al., 2016*); from this resulting dataset then retrieve the literature annotated with GO (*Ashburner et al., 2000*) terms like "chemical homeostasis" or any of its subclasses, e.g., "lipid homeostasis" and "triglyceride catabolic process" as well as the NCIT terms "insulin" and "insulin signaling pathway" as well as the the SNOMED term "homeostasis". While semantic annotations make it possible to define very specific queries, federated SPARQL makes it possible merge data distributed across the web. The researcher may also be interested in complementing the results with information from the Colil database (*Fujiwara & Yamamoto, 2015*). Colil searches for a cited paper in the Colil database and then returns a list of the citation contexts and relevant papers based on co-citations. The entire query is illustrated in Fig. 8 and the SPARQL code is available at *Biotea (2017h)*.

**Table 8  Queries against Biotea.**

| Queries | Federated Y/N | Ontologies | Endpoints |
|---|---|---|---|
| Get the title and the PMC identifier for articles annotated with Chemical homeostasis, including its subclasses or Insulin or Homeostasis as well as their COLIL citation context and the Insulin related pathways from Reactome | Y | SNOMED CT, GO, NCIT | Biotea, Reactome, COLIL |
| Retrieve all the articles containing Placebo Control, Crossover Study, Glucose tolerance test, Insulin secretion, glucose metabolic process and the entries from Uniprot related with glucose metabolic process, response to insulin and Diabetes mellitus, non-insulin-dependent (NIDDM) | Y | NCIT, SNOMED CT, GO, Uniprot | Biotea, Uniprot |
| Get all the annotations from GO and ChEBI in articles containing "American Joint Committee on Cancer" | N | GO, ChEBI, SNOMED CT | Biotea |
| Common SNOMED CT tags for articles pmc:3875424 and pmc:3933681 | N | SNOMED CT | Biotea |
| Get all the annotations for the article pmc:3865095 | N | Multiples vocabularies | Biotea |
| Get all the articles annotated with "Calcitocin" and "Injury of kidney" with it's PMC links and the DBPedia "Calcitocin" description as well as the Uniprot entries classified with "Calcitocin binding" | Y | Biotea, SNOMED CT, GO, Uniprot, DBPEDIA | Biotea, Uniprot, DBPedia |
| Retrieve all the articles annotated with "Renal cell carcinoma" and that cite them in the Open Citations dataset | Y | Open Citations | Biotea, Open Citations |

### Biotea and R

We also illustrate how to calculate the cosine similarity between pairs of papers with R, see *Biotea (2017a)*. In this example, we first retrieve all the articles annotated with SNOMEDCT:63736003 (Taiwanese), SNOMEDCT:110483000 (Tobacco user), SNOMEDCT:702675006 (Probe with target amplification) then, we calculate the Cosine Similarity between any pair of articles in the resulting dataset. The Cosine Similarity (*Jannach et al., 2010*; *Armstrong, 2013*) calculates the distance between two articles taking into account only the annotations in the documents. We visualize the results using a heatmap matrix; the darker the cell, the more similar the articles. Unlike the previous example, in this case we are only calculating the similarity; we are not using any clustering algorithm. The heat-map, see Fig. 9, illustrates the Cosine as a metric for semantic distance/similarity.

## DISCUSSION

We have generated linked data for PMC-OA; we are reusing existing ontologies for modeling the annotations, structure, metadata and the content in these documents. This new version of the dataset makes it possible for researchers to generate annotations using the AO or the OADM models; furthermore, annotations can now be generated from either XML or RDF files. The resulting dataset is over 150 Gigabytes in size and covers 7,407 journals. Our model uses domain ontologies that are widely used in biomedical databases; these databases have endpoints exposing their content as RDF and linked data. For instance,

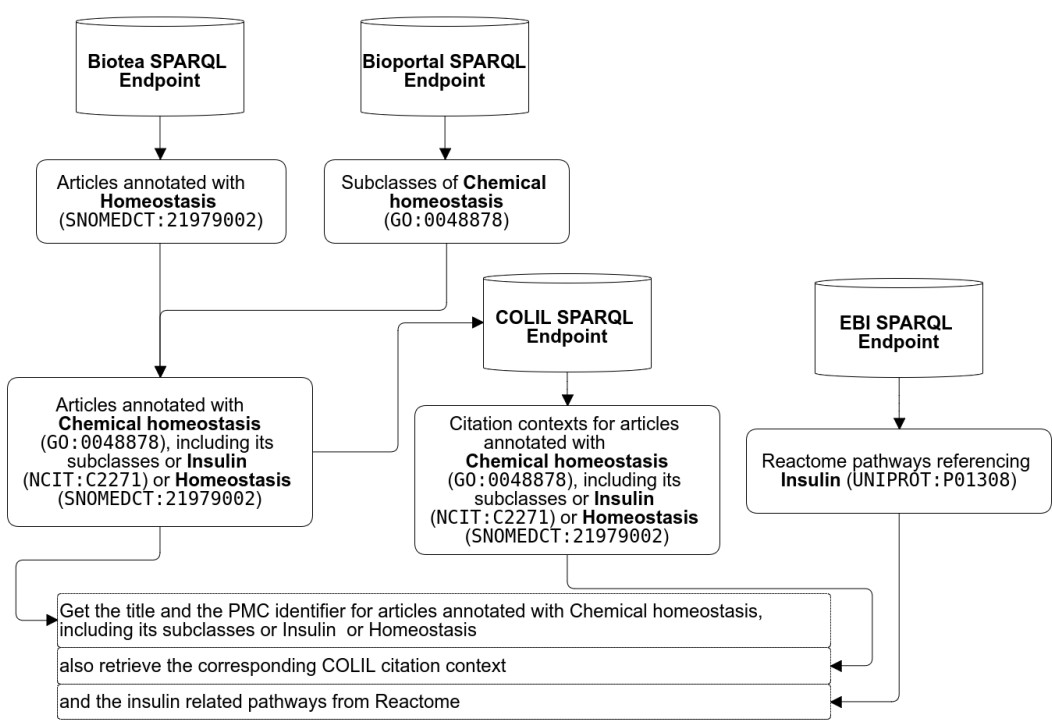

**Figure 8** Example of federated SPARQL Query.

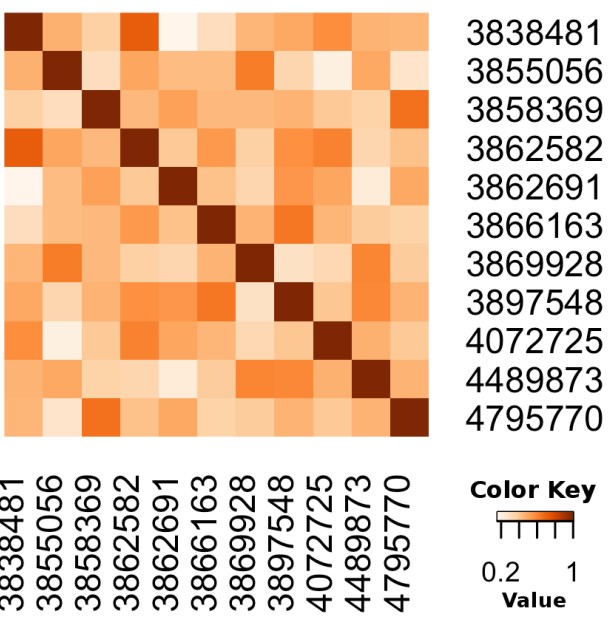

**Figure 9** Calculating the distance between pairs of articles using annotations.

the EBI RDF platform makes it possible for researchers to query across RDF datasets for resources such as Ensembl (*Aken et al., 2016*), BioModels (*Li et al., 2010*), Reactome, UniProt (*UniProt Consortium, 2017*), etc. The use of common vocabularies makes it easier to define the queries and thus relate information from heterogeneous sources via federated queries.

Our RDFization process is now more flexible as it has been divided into smaller tasks. This makes it easier for the of metadata, content and annotation to evolve independently as processes may be parallelized. Modularization also makes it easier to control the process; with more than one million documents to RDFize and annotate, managing the process is important. We have also added full support for the generation of Bio2RDF compliant outputs using the SIO ontology; it is possible to produce the RDF following the Biotea or the SIO compliant model or, both. Our mapping is not hard coded, it is expressed in a configurable file; this makes it easier for us to maintain the code independently from the changes on either model or simply adding new mappings to other models.

The availability of semantic annotations, the use of existing ontologies and, the RDFization of the content are key differences between SciGraph (*Springer Nature, 2017*) and Biotea. The SciGraph dataset makes use of a proprietary vocabulary; for interoperability purposes they also provide mappings to other vocabularies. The Biotea model is currently mapped to BIBO, DCTERMS, Dublin Core (DC), VIVO (*VIVO, 2017*), Publishing Requirements for Industry Standard Metadata (PRISM), as well as to other vocabularies. The one-to-one nature of these mappings imposes the same limitation as that described earlier for our dataset—see "Integration with Bio2RDF", last paragraph. Moreover, the use of different vocabularies to describe the same entity makes mapping based approaches expensive in terms of maintenance and flexibility. Whenever possible it is a good practice to reuse existing vocabularies instead of creating new ones. Furthermore, the SciGraph model is not as granular as that of Biotea; it models the journal and the paper but it does not addresses the content. In general, both data sets are compatible via the use of identifiers, e.g., DOIs. Our dataset complements that of SciGraph; for instance, the sg:subject (sg is the prefix for SciGraph) is defined as a "Subject" class that represents a topic. This is a field of study or research area that can be used to categorize the content of a publication; our annotations can be used to extend this class in the SciGraph dataset. Also, our dataset links to external resources and supports the representation of manual annotations. An interesting aspect in SciGraph is the use of sg:hasCrossrefFunderID for modeling funding information; this is an interesting addition that we are considering to reuse. Instances for "funders" may also come from repositories such as *OpenAIRE (2017)* and *SHARE (2017)*.

Our dataset is fairly sizeable; updating the dataset with only the most recent papers being added to the PMC collection was not initially addressed by our work. For this release we have tested the PubRunner (*Anekalla et al., 2017*) in order to periodically process only the most recent entries to PMC. In order to make it easier for us to release updates of the dataset we are modifying PubRunner and adapting it to our case. In this way we will be able to automatically focus on new data; thus, making it easier for us to manage the process and for consumers to use only the latest datasets. The size of our dataset is due to the verbosity implicit in the RDF/XML serialization. We are considering HDT

(Header, Dictionary, Triples) (*Fernández, 2012*) a solution for this problem; HDT is a compact data structure and binary serialization format for RDF that keeps big datasets compressed to save space while maintaining search and browse operations without prior decompression.

The Biotea dataset inherits the limitations from the annotations pipelines used to produce it—namely, the NER service provided by the NCBO. For instance, the disambiguation of "harbor" as a verb and "harbor" as a noun with a meaningful context from SNOMED (snomed:257621007) in a sentences like "direct sequencing of exons 5–8 which harbor 95% of the known..." poses a challenge to the NCBO annotation system. Generating lists with words that should be excluded from the annotation pipeline, e.g., stop words, is possible; the configuration file in Biotea (see *Biotea, 2017i*) makes it easy to generate such lists. Although we are following the best practices suggested by the NCBO annotator, using online services for such large datasets is not advisable. We had better results, less error due to communication problems and better performance, when we used a local appliance of the annotator.

Our choice of the NER service provided by Bioportal was influenced by the results presented by Funk et al. (*Funk et al., 2014*; *Jovanović & Bagheri, 2017*); the NCBO annotator, built upon MGREP, delivers good precision of matching compared to MetaMAP (*Aronson & Lang, 2010*; *U.S. National Library of Medicine, 2017b*). Also, the NCBO annotator delivers reliable programmatic access as well as a a virtual appliance that can run locally with very little effort; moreover, as the single entry point for most biomedical ontologies, the NCBO annotator makes it unnecessary to search and install, with the consequent reformatting and parsing, ontologies and vocabularies. In addition, the NCBO annotator is very well supported; not only with extensive documentation but also with a community that facilitates the problem solving process. In this release of the dataset we didn't consider Machine Learinig (ML) methods. For our task, annotating the open access full text subset of PMC with several ontologies, there are no comprehensive datasets that can be used to train the models; existing annotated corpora focus on specific annotation targets -e.g., drug-drug, protein-protein interactions, identification of diseases, etc.

The current version of Biotea was not annotated with Whatizit (*Rebholz-Schuhmann et al., 2008*) because it is no longer available. This limits the knowledge encoded in our annotations as we are missing WhatIzit annotations pipelines such as those for UMLS diseases and UniProtKB proteins. These workflows were giving us direct links to databases such as UniProt. Some of these direct links are, however, resolvable, simply by using the endpoints available for the corresponding databases. For instance, "insulin" is currently linked to `PR:000009054` in the Protein Ontology (PR) while via Whatizit it would have been related to UniProtKB proteins such as `up:P01308` (INS_HUMAN), `up:P01317` (INS_BOVIN) and `up:P67970` (INS_CHICKEN). We can reach some of those links by getting the direct children of PR:000009054 which includes `PR:P01308` and `PR:P67970`; both of them are linked to UniProtKB proteins by means of the PR property `database_cross_reference`. On a different scenario, if we are interested in "high-density lipoprotein", Whatizit would have associated this term to proteins such as up:Q9D1N2 and up:Q8IV16. We are exploring different alternatives so the missing annotations, w.r.t.

the first Biotea dataset, can be automatically added. The RESTful web services available at EuropePMC (*Europe PMC, 2017*) make it possible to retrieve most of the annotations we were getting from Wahtizit, we are working on methods that allow us to use these annotations. The problem is that our model anchors the annotations to sections within the document whilst for EuropePMC these annotations are part of the document as a whole. We are evaluating Neji (*BMD Software, 2016*) and EuropePMC RESTful services as possible alternatives for replacing Wahtizit.

## CONCLUSIONS

By delivering a semantic dataset for PMC-OA we are making it easier for agents in the web to process biomedical literature. Having entities semantically characterized makes it possible for software agents to process them in various ways, e.g., using the association diseases-populations-interventions in order to link to health records or, by using the association gene-protein-disease to link to metabolic pathways. We are also making it possible for researchers to express queries using ontological concepts; these queries can be expanded against federated linked data resources in the web—hence improving recall. Semantic annotations are highly structured digital marginalia; these are usually invisible in the human-readable part of the content. In Biotea, annotations are represented using a machine-interpretable formalism. As illustrated in the prototype, notes are then used for classifying, linking, interfacing, searching and filtering.

Our approach is useful for both open and non-open access datasets; since the content is clearly identified and enriched with specialized vocabularies, publishers may decide what to expose as linked data. For instance, annotations may be published while the content may be kept hidden; in this way the benefits of conceptual queries could be made available over a SPARQL endpoint without compromising the content of the document. Having self describing documents, as we propose in this paper, also makes it easier to establish comparisons across documents; these should go beyond what we currently make possible. For instance, if tables were dynamically generated from semantically annotated data then researchers could easily establish comparisons across datasets reported in the literature. Such comparisons could also include annotations from one or more ontologies; in this way it could be possible to discern the differences and similarities with respect to, for instance, GO annotations. Selfdescriptive documents could also enrich the user experience when searching and interacting with the document, as it is suggested in our prototype as well as in our earlier experiments (*Garcia Castro, McLaughlin & Garcia, 2013*).

The Biotea dataset will continue to grow by adding new sources of annotations for our corpus. We will focus on maintaining Biotea as a resource where researchers are able to find annotations for biomedical literature: full content, open access. Annotation pipelines and NER systems will always have advantages and disadvantages with respect to each other; by having annotations under one roof, the Biotea data set simplifies the process of benchmarking and using annotations for particular purposes. Our next release will include annotations from the Whatizit pipelines as well as disease-gene associations from (*Pletscher-Frankild et al., 2015*). By adding new annotations, we will also improve the

quality and quantity of links between the content and web based information resources. Enhanced associations between genes, proteins and specialized databases will also be the focus of our next release. In our next release we will also continue exploring the use of annotations in supporting better user experiences; we will focus on query composition and data exploration.

### Funding

Olga Ximena Giraldo has been funded by the EU project DrInventor FP7-ICT-2013.8.1. Alexander Garcia has been funded by the KOPAR project, H2020-MSCA-IF-2014, Grant Agreement nr: 655009. Federico Lopez has been funded by linkingdata.io. The funders had no role in study design, data collection and analysis, decision to publish, or preparation of the manuscript.

### Grant Disclosures

The following grant information was disclosed by the authors:
EU project DrInventor: FP7-ICT-2013.8.1.
KOPAR project, H2020-MSCA-IF-2014: 655009.
linkingdata.io.

### Competing Interests

The authors declare there are no competing interests.

### Author Contributions

- Alexander Garcia conceived and designed the experiments, performed the experiments, analyzed the data, wrote the paper, prepared figures and/or tables, reviewed drafts of the paper.
- Federico Lopez and Leyla Garcia performed the experiments, contributed reagents/materials/analysis tools, prepared figures and/or tables, reviewed drafts of the paper.
- Olga Giraldo performed the experiments, analyzed the data, contributed reagents/materials/analysis tools, prepared figures and/or tables, reviewed drafts of the paper.
- Victor Bucheli reviewed drafts of the paper, supervised implementation and provided useful discussion.
- Michel Dumontier contributed reagents/materials/analysis tools, reviewed drafts of the paper.

### Data Availability

Biotea: http://biotea.github.io/; https://zenodo.org/communities/biotea/?page=1&size=20.

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
