# Peer review of "Biotea: semantics for Pubmed Central"

_PeerJ, doi:10.7717/peerj.4201_

## Round 0.1 · original submission · Minor Revisions

Please address all comments raised by the reviewers. Please add two more examples as suggested by the third reviewer.

Reviewer 1 ·

Basic reporting

I enjoyed reading the manuscript. The presented work is sound and highly practical. The manuscript is well written.

I commend the authors for semantically processing articles from >7K journals and making all the data and software code available. In my opinion the submitted manuscript should be accepted for a publication after suggested minor revisions:

- ‘the NER service provided by  the NCBO Annotator’ The accuracy of many other NER tools is higher than of this one, see for example tools provided by NaCTeM. Those other tools also are using ontologies. Please explain your choice.
- Overall, there is no discussion of text mining (TM) as a closely related area of research. Of course, there are principle differences with what the authors are producing, but many TM steps can be re-used by Biotea.
- I would prefer to see more examples and discussion o how Biotea can be used.
- Line 64: the first mentioning of Biotea – you need to provide more explanation what it is. The same about hypothes.is
- Proofread the manuscript, examples: Lines 48-49: add missing gaps; Line 65: “representation, idem. that “, etc. etc.

Experimental design

no comment

Validity of the findings

no comment

Additional comments

no comment

Reviewer 2 ·

Basic reporting

The authors do an excellent job of explaining what they mean by their term "RDFize". However, I do not see the need to coin a new verb, especially considering the authors later turn the this new verb into a noun: RDFization. I think it best editorial practice if they refrain from coining a new term and instead refer to RDF generation or RDF creation as is appropriate.

Suggested grammatical changes:

Lines 280-281. "We are using hierarchical ... using the cosine distance as the metric"
Line 409. "The resulting dataset is over 150 Gigabytes in size"
Line 421. Parameterize (actually suggest rewriting this sentence - it's not completely clear)

Experimental design

No comment

Validity of the findings

No comment

Additional comments

Just a suggestion. You note several software dependencies for using Biotea - Maven, Java and Eclipse. These particular programs tend to have very specific versions for various OS platforms. I think it would be helpful to your audience and potentially increase the usage of Biotea if you were to provide a preconfigured Virtual Machine image using Ubuntu. Virtualbox provides such customized VM's for various purposes (http://www.oracle.com/technetwork/community/developer-vm/index.html), as well as Bitcurator (https://www.bitcurator.net/)

Reviewer 3 ·

Basic reporting

The subject of the manuscript is one that is important to the future of science, an one that is close to my heart: the improved reporting of scientific knowledge through the use of semantic technology. There is a clear and urgent need to improve the reporting of scientific knowledge, and it is scandalous that so much public and charitable money is spent on science that cannot properly be used because it is inaccessible to computers.

The Introduction and background are clear, and the literature is well referenced and relevant. Clear, unambiguous, professional English language is used throughout. The figures are relevant, high quality, well labelled and described.

Experimental design

The models, services, software and datasets are available.

Validity of the findings

The authors demonstrate the utility of Biotea in two examples. This is the weakest part of the manuscript:
• Example 1 concerns the retrieval and clustering of papers annotated with the SNOMED CT term ‘renal 
cell carcinoma’. Unfortunately, the three papers investigated have little to do with ‘renal 
cell carcinoma’, although it is true that this phrase occurs in all of them. However, the authors do a good job of describing why papers 3862691 and 3862582 are more similar with each other than with 3899087.
• Example 2 involves the creation of a very long SPARQL query, but the results of the query are not described. The SPARQL query would be very hard for a domain scientist to generate without the use of some tool.
I recommend that two other examples are used to demonstrate the utility of Biotea.

Around line 255 the manuscript describes the mapping between Biotea and SIO concepts: ‘encapsulating the original data type property value; thus, a bibo:pmid with the value “28300141” is 
mapped to the object property sio:has_identifier, this is linked to the class sio:identifier 
that is related by means of sio:has_value to the actual PMID “28300141”.’ I don’t see how this mapping captures the key information that the identifier comes from PubMed .

---

## Round 0.2 · accepted · Accept

You have addressed all of the reviewers' comments. Your manuscript has been accepted for publication.